# The Role of Protein Degradation in Estimation Postmortem Interval and Confirmation of Cause of Death in Forensic Pathology: A Literature Review

**DOI:** 10.3390/ijms25031659

**Published:** 2024-01-29

**Authors:** Weisheng Huang, Shuquan Zhao, Huine Liu, Meichen Pan, Hongmei Dong

**Affiliations:** 1Department of Forensic Medicine, Tongji Medical College, Huazhong University of Science and Technology, No. 13 Hangkong Road, Hankou, Wuhan 430030, China; weishenghehe@163.com (W.H.);; 2Faculty of Forensic Pathology, Zhongshan School of Medicine, Sun Yat-sen University, Guangzhou 510080, China; zhaoshq27@mail.sysu.edu.cn

**Keywords:** protein technology, postmortem protein degradation, the cause of death, postmortem interval

## Abstract

It is well known that proteins are important bio-macromolecules in human organisms, and numerous proteins are widely used in the clinical practice, whereas their application in forensic science is currently limited. This limitation is mainly attributed to the postmortem degradation of targeted proteins, which can significantly impact final conclusions. In the last decade, numerous methods have been established to detect the protein from a forensic perspective, and some of the postmortem proteins have been applied in forensic practice. To better understand the emerging issues and challenges in postmortem proteins, we have reviewed the current application of protein technologies at postmortem in forensic practice. Meanwhile, we discuss the application of proteins in identifying the cause of death, and postmortem interval (PMI). Finally, we highlight the interpretability and limitations of postmortem protein challenges. We believe that utilizing the multi-omics method can enhance the comprehensiveness of applying proteins in forensic practice.

## 1. Introduction

Proteins are the basic substances of the human life process, and they play an extremely important role in both physiological and pathological activities. The function of proteins is dependent on the diversity in both the specific amino acid sequences and the spatial structure of the proteins [1]. The detection of enzyme activity and the changes in protein content can serve as biomarkers for clinical diagnosis and disease monitoring. For example, elevated serum lactate dehydrogenase content levels can be a marker for myocardial infarction [2]; increased serum alpha-fetoprotein levels indicates early hepatoma; and alanine aminotransferase and aspartate aminotransferase are indicators of liver function impairment in clinical practice. Proteins or polypeptides can also serve as effective drugs in the clinic, such as insulin, human gamma globulin, and some enzyme productions.

In forensic practice, the traditional morphological technique remains a primary method for medicolegal expertise. Although difficult cases may arise sometimes, the specific stains of the target biomarkers can still help forensic pathologists to ascertain the cause of death in judicial autopsy. However, challenges have always existed when forensic pathologists encounter ‘negative autopsies’, which refers to corpses without any distinct characteristic morphological findings, such as diabetic ketoacidosis (DKA), diabetes, arrhythmia, heatstroke, etc. The changes in targeted proteins are thought to have potential practical value in dysfunction-related deaths and other challenging cases. Skin is a good biological sample in cadavers, capable of detecting DNA typing with specific stains of nucleus chromatin (Lillie’s staining) even in leathery and mummified skin samples [3]. Sun et al. [4] utilized the P/VB-specific staining of collagen and muscle near the scalp in a decomposed cadaver, and confirmed the fatal antemortem wound of the decedent. However, the histological methods may not address various conditions in an advanced decomposed cadaver. In other words, the limitations of the histological method hinder the potency of proteins in estimating the PMI of decomposed corpses or confirming the cause of death in ‘negative autopsies’. With the development of technology, novel techniques and methods, such as proteomics and metabolomics, have been attempted in forensic practice to obtain much more information about the body’s pathophysiological process. In recent years, the proteomics approach has been attempted in body fluid identification, sexual identification, biomarkers for species identification, peptide toxicology, and the estimation of postmortem interval (PMI), etc. [5]. And the fourier transform infrared spectroscopy (FT-IR) and metabolomics have been used to identify the subtle changes in ‘negative autopsies’ in forensic practice.

## 2. Metarials and Methods

The literature review was carried out using PubMed NCBI databases. The papers published in last 20 years range from 2003 to 2023. This review is intended to discuss the relevant papers that focused on clarifying the rule of protein degradation applied in confirming causes of death and estimating PMI. Several key terms were utilized in searching relevant papers, due to multiple protein techniques being applied in forensic practice. In this review, the following key words were used: “(immunohistochemical staining) AND (forensic science)”, “(((postmortem interval) OR (cause of death)) OR (injury))) AND (western blot) AND (forensic science)”, “((postmortem interval) OR (cause of death)) OR (injury))) AND (ELISA) AND (forensic science)”, “((postmortem interval) OR (cause of death)) OR (injury)) AND (Proteomics)) AND (forensic science)”, “((postmortem interval) OR (cause of death)) OR (injury)) AND (Metabolomics)) AND (forensic science)” “((postmortem interval) OR (cause of death)) OR (injury))) AND (Fourier transform infrared spectroscopy) AND (forensic science)”, and “(Peptidomics) AND (forensic science)”.

The articles were selected when they met with the following criteria:Quantitative or qualitative postmortem evaluation of proteins or its metabolites on animal or human tissues;English language;Year of publication from 2003 to 2023.

Articles were evaluated via paper title, abstract, and content in accordance with the agreement of the works with our inclusion criteria. A total of 137 papers were included in this review (Figure 1). The inclusion methodology was handled in strict accordance with the guidelines of the PRISMA.

## 3. Results 

The inclusions of keywords in the PubMed NCBI provided 1014 relevant articles. In Figure 1, the flow of articles retrieved for the review is reported. Papers published before 2003, duplicated titles, and non-English works were excluded. In total, 347 works were screened, of which, 217 were chosen for full text reading and 137 papers were included in this review. The selected papers included forensic researchers assessing postmortem protein detection with existing means to estimate PMI and select the biomarkers related to the cause of death. 

### 3.1. Immunohistochemistry (IHC) Assay

IHC is a classical and routine diagnostic method in pathology. In forensic practice, IHC also serves as a common supplementary method in diagnosing the cause of death, confirming the causality, and estimating PMI, etc. (Table 1). 

As shown in Table 1, the IHC method has been applied in almost all the hot issues in forensic pathology. In determining the cause of death, IHC assays play an important role in postmortem diagnosis in forensic pathology. They greatly reduce the diagnosis difficulties, due to their being sensitive to specific proteins related to suspicious fatal injury or disease. Single immunochemistry staining was applied in diagnosing specific trauma due to its sensitivity in identifying the target protein, such as the IHC staining of tubular myoglobin aiding in the verification of crush syndrome in judicial autopsy [37]. The positive expression of CD31 and cytokeratins in fetal lung tissue can help forensic pathologists definitely diagnose amniotic fluid embolism more easily compared with routine hematoxylin eosin detection [30]. It was known that morphological features are often absent in some negative autopsies. The IHC staining of exogenous fatal protein is also crucial in confirming the cause of death. Tong et al. [38] utilized IHC to detect the insulin adjuvant-protamine near a suspicious injection site, thereby helping to confirm insulin overdose. Wisnewski et al. [39] used immunostaining for methylene diphenyl diisocyanate at the basement membrane of highly inflamed bronchioles to demonstrate that fatal severe asthma may occur in workers with long-term exposure to methylene diphenyl diisocyanate in the workplace. In some cases of sudden death (Table 1), the presence of different types of inflammatory cells serve as the specific pathological features, reflecting the inflammatory response before the decedent’s death. For instance, anti-tryptase antibodies can ascertain anaphylactic death through labeled eosinophils, and CD68 can clarify the injury age using labeled macrophages, etc. Heat shock proteins (HSP) can assist in maintaining cellular homeostasis due to their functions in various physiological and protective processes. Forensic pathologists have utilized different variations in HSPs under diverse stressors (including hypoxia, and thermal stress), designating HSP27 and HSP70 as target proteins for the postmortem diagnosis of heat stroke, death in fire, and hypothermia (Table 1). Specific IHC stains can also help forensic pathologists to determine the final cause of death when they encounter sudden death in patients with systematic immune disorders [40]. With the application of IHC in complex forensic judicial autopsies, more and more cases achieve a precise diagnosis of the cause of death. Dettmeyer et al. [41] utilized the IHC method to confirm the diagnosis of cytomegalovirus-induced pneumonia and myocarditis in three cases of suspected sudden infant death syndrome. Chen X et al. [42] utilized IHC to stain myocardial connexin 43, and found that the protein expression was reduced in a decedent with dilated cardiomyopathy. Forensic pathologists utilized multiple immunohistochemical stains as a tool to present more persuasive evidence. For example, acute myocardium infarction, a common fatal disease in forensic practice, may not exhibit distinguishable histological changes in both coronary vascular and myocardium sometimes, especially when it occurs in a short period as short as 1 h. Mondello et al. [11] reviewed a total of 27 related articles, and concluded that immunohistochemical stains of specific proteins provide a substantial contribution to determine the stage of myocardial infarction.

After the outbreak of the coronavirus disease (COVID-19) epidemic, the autopsy reports of COVID-19 victims were promptly available online, which presented a great help to the modification of treatment guidelines in China. And IHC is an important pathological tool to explore the pathogenic mechanism [43]. In forensic practice, IHC also provides direct evidence of virus infections, and clarifies the causality between virus infection and death. Živković et al. [44] performed an autopsy on a 57-year-old female infected with COVID-19, and diagnosed that she died from Guillain–Barré syndrome secondary to SARS-CoV-2 infection, dependent on the IHC staining of CD68 and CD3. Respiratory syncytial virus (RSV) infection is the most common cause of lower respiratory infection, which is fatal in children less than 1 year old. Ikeda et al. [45] proposed that RSV immunopositivity is located in the bronchial epithelium and intra-alveolar cells, which helps to determine the cause of deaths related to RSV infection. In our previous study, we presented a rare case of acute necrotizing encephalopathy based on the influenza A virus NS1 immunopositive located in the main organs (including bilateral thalamus and pons, epithelium of the larynx, and the pulmonary alveoli) [46].

Additionally, the IHC detection of antigens may provide a rough estimation of PMI in some single cases. Some researchers have attempted to utilize the gray value of IHC staining to evaluate PMIs less than 48 h [47,48].

In general, as shown in Table 1, the IHC method could help to confirm actual causes of deaths due to its high sensitivity to a series of target proteins. It can serve as an integral part of supplementary evidence to establish the final diagnosis. Although immunohistochemical staining may not exhibit unique features, in some cases, it can still help to provide objective evidence to investigate the truth of judicial cases. It can also help forensic practitioners to identify a rough estimation of the PMI in the casework [49]. However, the utilization of IHC is still limited in forensic pathology, primarily due to the protein degradation of the corpses, which can affect the selected biomarker and the IHC technology.

### 3.2. Enzyme-Linked Immunosorbent Assay (ELISA) and Western Blot

Forensic molecular pathology approaches are effective means in routine forensic jobs, major education, and practical training [50]. For protein detection, ELISA and Western blot are two potent methods.

The ELISA method has been applied in forensic practice to detect and quantify a series of organic substances (such as protein, polypeptide, bioamine, etc.) from body fluids or tissue homogenates (Table 2). 

Forensic researchers have exploited a series ELISA method, which is applied in forensic practice to confirm suspicious toxins. Huang et al. [51] utilized ELISA to detect Deinagkistrodon acutus venom in multiple organs, including the kidneys and liver tissues, as well as a suspicious bite injury in the left thigh muscle of a victim suspected to be bitten by a Hubei province native poisonous snake. ELISA can be used to assess the postmortem redistribution of suspicious toxins and select the potential predictors for cardiotoxicity or neurotoxicity (caused by drugs, such as amphetamine, methamphetamine, and ketamine) [52,53,54].

ELISA has mainly been used to select postmortem protein indicators for determining traumatic brain injury (TBI) and early myocardial ischemic-induced sudden cardiac death, etc. (Table 2). Forensic practitioners have made multiple attempts to select the biomarkers related to TBIs and acute myocardial infarction using the ELISA method.

Shang et al. [66] utilized the ELISA approach to analyze the serum levels of neurogranin (NG) and myelin basic protein (MBP) in postmortem blood samples from 57 decedents caused by traumatic brain injury, which was used to research the postmortem changes of these two indicators. The research indicated that an ELISA analysis of the serum levels of NG and MBP might be a novel method to diagnose TBI in humans, with further validation of IHC and immunofluorescence stains in the injury area of the brain. The postmortem detection of serum tenascin-C, IL-6, and IL-8 can help forensic pathologists to determine the causes of fatal traumas in TBI autopsies [67,68]. In addition, researchers have utilized the ELISA method to overcome challenges in forensic practice, such as the cause of death in ‘negative autopsies’ and PMI estimation. Panwar R. et al. [69] employed the ELISA method to identify TNF-α and IL-3 as two reliable biomarkers for detecting mast cells, which could increase under a prolonged hypoxia condition. And this optimization method can help forensic pathologists to differentiate fatal asphyxia decedents easily from other causes of death.

Haynes et al. [63] proposed that high serum 5-HT may be a characteristic feature of SIDS. Kasuda et al. [64] thought that urinalysis may help to estimate PMI depending on VWF detection using ELISA. Zhang et al. [16] utilized the ELISA method to detect biomarkers of cantharidin-induced myocardial damage, which indicated that the plasma levels of TN-T and HIF-1α/TN-T can help to definite the cardiotoxicity, and the value of VEGF/HIF-1α has a strong association with PMI within 168 h. However, the study of PMI estimation with ELISA needs a lot of samples to validate, due to irregular variation in postmortem body fluids (for example, the autolysis of blood, etc.).

Western bolt is another important method for quantifying protein expression, which is commonly used to research the biomarkers of wound age and PMI in complex cases. The vital reaction near the wound is critical for distinguishing between a wound sustained while alive and a postmortem wound. Kimura et al. [70] utilized Western blot technology to detect the autophagy-related factors (LC3-II and p62), which have significantly different expressions in wound samples in decomposed cadavers (PMI: 1–4 day) and are the potential novel biomarkers for wound vitality in cadavers.

The postmortem degradation rule of protein may provide the possibility of precisely estimating PMI in decomposed corpses. Sabucedo et al. [71] utilized Western blot assay to analyze the degradation rule of cTnI, and found that cTnI could serve as a biomarker in early PMI (0–5 days). Pitter. S. et al. [72] used Western blot to analyze the protein degradation of porcine biceps femoris muscles 10 days after the pig slaughter. They selected several myofibrillar structural proteins (titin, nebulin, desmin, cTnT, etc) as novel biomarkers in estimating PMI. Subsequently, Pitter’s research team validated the muscular protein degradation rule of muscular tissue in other mammal and human samples, and found the same degradation of the desmin in mammal’s muscle tissue [73]. In forensic autopsy, different types of muscle samples have a highly conserved process, making it a reliable method with strong validity and replicability [74] The expression of cardiac troponin T (cTnT) in the myocardium could also help to estimate PMI, as it exhibits a pseudo-linear relationship between the log value of the PMI and the degeneration of cTnT [75].

However, the entire proteome will change its contents after an animal dies. Thus, researchers did not set the control protein as a reference protein, making it difficult to evaluate the real degradation trend. Lee et al. [76] analyzed the half-life period of eight target proteins’ degradation in kidney and skeletal muscle tissue, integrating the methods of ELISA, IHC, and Western blot. They found that glyceraldehyde3-phosphate dehydrogenase (GAPDH) emerged as a mid-term PMI marker, while p53 and β-catenin remained constant for at least 96 h since death. However, it should be noted that the whole proteome begins to degrade at postmortem, rendering the routine internal reference protein (GAPDH, or β-actin) unsuitable for the relative quantification of target proteins in Western blots. Therefore, in previous studies, researchers used the protein content of a 0 h group as reference to evaluate the degradation of the target protein depending on its percentage contents in different time points after death.

Both ELISA and Western blot assays possess higher sensitivities to target proteins, making it easier to detect and quantify the target protein compared with the IHC method. The ELISA method is usually used to evaluate the degeneration of proteins caused by stressors in body fluid (Table 3). The Western blot assay can be used to evaluate the postmortem protein degradation of relevant tissues, presenting a valid means for PMI estimation. Thus, both ELISA and Western blot methods play an important role in gathering evidence. However, forensic pathologists might need to repeat detection in some cases without clues, due to their poor coverage of target substance (Table 3).

### 3.3. Proteomics

In clinical medicine, proteomics has been widely utilized in clinical studies to screen target biomarkers, select vaccine candidates, study the mechanism of serious diseases, and confirm signal pathways [77]. During the COVID-19 epidemic in 2020, Chinese researchers utilized a Tandem Mass Tags (TMT)-based liquid chromatography tandem mass spectrometry (LC-MS/MS) analysis to observe seven main organs (including the heart, the liver, the spleen, the kidney, the thyroid, and the testis) from 19 COVID-19 decedents [78], which indicated 5336 abnormal expression proteins out of 11,394 proteins, providing a comprehensive understanding of the infectious mechanism in each organ.

Forensic science focuses on clarifying the phenomenon after death. The degradation of protein is a complex process in corpses, which is influenced by a series of biotic (microbe and parasite insects) and abiotic (temperature and pH value) factors. Therefore, the routine forensic molecular pathology approach may not provide a comprehensive understanding of the general profile of protein degradation after death through the observation of specific target proteins. With the development of the proteomics approach, a bottom-up in situ proteomics method was exploited in detecting kinds of samples, which fits in forensic pathology research and practice [79]. Wilke [80] proposed that the proteomics method could play an important role in presenting massive data to provide effective evidence in forensic challenges, such as confirming protein toxins, PMI estimation, and identifying special tissues or biofluids.

In forensic practice, cases involving fatal protein toxins (such as insulin overdose, snake venom, etc.) occasionally occur. However, forensic pathologists cannot use the common method to qualify or quantify protein toxin sometimes due to the unremarkable wounds on the surface of the body and the postmortem degradation of the proteins. Beckett et al. [81] used the proteomics approach to detect insulin analogues in the postmortem vitreous humor of the decedent, who applied an overdose insulin before his death. Chan et al. [82] utilized proteomics to find the specialized protein ‘phospholipase A2’ of snake venom in the victim’s blood samples, which was contributed to confirm the actual cause of death.

In judicial autopsies, forensic pathologists should provide the objective pathological evidence and explain its mechanism of death for the decedents’ family. In clarifying the pathogenesis, forensic researchers have conducted a lot of studies on traumatic brain injury, cardiac diseases, and toxin attacks on specific organs in mammal animals. Sudden cerebral death and sudden cardiac death are two hot issues in forensic pathology due to their high fatality. TBI can lead to a severe axonal injury in both the cerebral injury area and its peripheral cerebral tissue. In confirming the target biomarkers of diffuse axonal injuries, Zhang et al. [83] selected 10 differentially expressed proteins (DEPs) as the diffuse axonal injury (DAI) biomarkers in rat cerebral tissue, with isobaric tags for relative and absolute quantification (iTRAQ) coupled with LS-MS/MS analysis. Coincidently, our previous study selected 14 DEPs in the corpus callosum with the same proteomics method, involving calcium ion-regulatory proteins, cytoskeleton organization, etc [84]. Chen et al. [85] utilized TMT-based LC-MS/MS to analyze the potential biomarkers of DAI in brain stem, which indicated that CRB-1, EPHX2, and CYP2U1 serve as novel biomarkers of DAI. In cases of cardiac death, the myocardium may be damaged by cardiac disease, ischemic cardiac disease, and alcohol dependence. Kakimoto et al. [86] utilized shotgun proteomics to analyze the mechanism of death in acquired cardiac hypertrophy autopsy, and indicated that the stepwise upregulation of sarcomere protein is a danger signal ACH, which may help to diagnose the sudden cardiac death of ACH. Ma et al. [87] selected 56 DEPs with TMT-based LC-MS/MS, and proposed that the chronic ethanol exposure could lead to hypertrophy of cardiomyocytes, and the dilation of the ventricle, etc.

Additionally, proteomics also has an obvious advantage in detecting postmortem DEPs for PMI estimation too. Zhang et al. [88] utilized the two-dimensional electrophoresis to analyze protein degradation in the liver and spleen of the rats, which indicated that β-actin has a strong association with PMI, and SBP2, ENOA, and ALDH2 can be serve as internal references in detecting the protein degradation. Skeletal muscle is a common sample that is used to estimate early PMI. Choi et al. [89] utilized proteomics to detect PMI-related DEPs in rat skeletal muscle, and identified GAPDH and eEF1A2 as potential PMI biomarkers within 96 h after death. Battistini et al. [90] utilized proteomics method to analyze the skeletal muscle of pigs, and proposed that three validation proteins (PDLIM7, TPM1, and ATP2A2) serve as PMI biomarkers. Mizukami et al. [91] analyzed the skeletal muscles and skeletons from the aquatic decomposition of rats, demonstrating the potential of proteomic to estimate PMI in an aquatic environment. Brockbals et al. [92] utilized the proteomics method to analyze human skeletal tissue in different seasons, and proposed the multiple peptide ratio for PMI estimation. Skeletons are the optimal samples to evaluate late PMI, particularly one in excess of several years, in forensic investigation [93,94]. Researchers have identified 32 DEPs as PMI biomarkers in human skeletons, which can be used to verify PMI for more than 10 years [95]. Peptides, the products of postmortem protein degradation, also have the potential to estimate PMI. Bonicelli et al. [96] found 10 DEPs in human skeletons from casework, serving as the most interesting biomarkers.

In addition, proteomics has been applied in casework, Dammeier et al. [97] utilized the forensic proteomic method to detect the proteins on the surface of a bullet after shooting, which could help to confirm fatal gunshot wound. To date, proteomics has been generally applied in forensic science to overcome major difficulties. However, it still cannot be regarded as a common method now due to the high costs (Table 1).

### 3.4. Peptidomics

As the ‘bridge’ between proteomics and metabolomics, peptidomics is also an effective method to quantify endogenous peptides. Clinical researchers usually utilize an untargeted peptidomics method to select biomarker candidates, which are validated with targeted peptides.

In forensic science practice, peptidomics is considered a new-generation serological technology based on the protein mass spectrometry for seminal fluid detection. And the peptidomics method has been utilized to identify the sex of the decomposed corpse and confirm the type of the suspicious biofluid. Ziganshin et al. [98] utilized peptidomic analysis to confirm the sex based on the distinct contents of the amelogenin Y isoform, and proposed that the phosphorylated Ser66 residue was the specific substance in the enamel from deciduous teeth. This method can help forensic practitioners to confirm the sex of juvenile remains or adult skeletons in advanced decomposed preservation. Forensic researchers have utilized the peptidomics approach to detect specific peptides (Semenogelin, Prostate Specific Antigen/p30, and Prostatic Acid Phosphatase) to confirm seminal fluid, aiding in the verification of sexual assault cases [99].

In addition, peptides, as the basic components of protein, are potential PMI biomarkers. Nolan et al. [100,101] utilized high performance liquid chromatography-triple quadrupole mass spectrometry to select 29 peptides for differentiating PMI stages in pig decomposition fluid.

### 3.5. Metabolomics

Metabolomics is not only a simple biomarker identification tool but also a crucial tool for precision medicine [102]. In clinical medicine, metabolomics has been applied to various chronic diseases, including cancer [103], heatstroke [104], and diabetes [105], etc. In forensic science, metabolomics has also been used to overcome forensic difficulties, including cause of death in complex cases, toxicological features in target organ, and PMI estimation.

In the research on complex causes of death research, metabolomics serves as a novel technology to identify the potential biomarkers of difficulties in forensic practice. Wang et al. [106] established a metabolomics based on gas chromatography mass spectrometry (GC-MS), and found that valine, octadecanoic acid, and leucine are strongly associated with acute ischemic stroke. Asphyxia is a difficult issue in forensic pathology. Asphyxia ultimately leads to cardiac arrest, making it difficult to distinguish from primary cardiac arrest due to the absence of specific biomarkers. Zhang et al. [107] utilized the untargeted metabolomics to discover novel evidence for the differentiation of asphyxia from sudden cardiac death in decomposed corpses of experimental animals, which is acquired from the profile of lactic acid, pantothenic acid, and eight other kinds of potential metabolites. It is difficult to determine the cause of death in judicial autopsy, when forensic pathologists suspect the cause of death to be diabetes mellitus and its complications, due to the scarcity of biomarkers. Nariai et al. [108] used the metabolomics approach to select biomarkers related to diabetes mellitus after death, and found a group of metabolites to diagnose diabetes mellitus, including sphingomyelin, plasmalogen lipid, and lysophospholipid. In postmortem diagnosis, Zhang et al. [109] selected 36 metabolite biomarkers to develop the classification model, which can help forensic pathologists in PMI estimation and the identification of drowning and postmortem submersion.

The toxicometabolomics method can reveal toxin(drug)-induced multi-organ injuries [110]. A lot of studies focus on the hepatic injury caused by methamphetamine [111], heroin [112], cantharidin [113], alcohol [114], arsenic [115], etc. In addition, metabolomics has also been used to select the biomarkers related to some drug overdose. Ward LJ et al. [116] utilized the metabolomics method to identify 12 acylcarnitines relevant to hypoglycemia-related death in autopsy. Dai X et al. [117] selected three endogenous metabolites (phenylacetylglycine, creatine, and indole-3-lactic acid) in serum and three metabolites (palmitic acid, creatine, and indole-3-lactic acid), respectively, establishing a characteristic metabolomics profile for estazolam fatal intoxication.

After human death, proteins undergo initial breakdown into long chain peptides and amino acids due to the action of proteases and endopeptidases. Various free amino acids undergo processes including deamination, hydroxylation, hydrolysis, and redox reactions, leading to the generation of amines, ammonia, hydrogen sulfide, dioxide, hydrogen, and phenol, etc. Thus, metabolomics could serve as a potential method to elucidate the process of protein bio-degradation process. After human death, the corpse undergoes decomposition over time. With the extension of PMI, the biodegradation pathways of organics (including three types: lipids, proteins, and carbohydrates) lead to their basic unit, and the amino acids and amine particularly present in decomposed fluid [118]. In 2005, Statheropoulos et al. [119] pioneered the application of metabolomics in postmortem changes by using this method to detect over 80 kinds of small volatility molecular substances in a drowned body from ocean water for 3–4 weeks. The researchers proposed that this approach holds potential as a precise diagnostic tool for estimating PMI and determining the cause of death in decomposing corpses. Recently, metabolomics has been applied to detect the changes in soft tissue and bio-fluid, aiding the evaluation of the degradation of organic substances. Szeremeta et al. [120] found 51 metabolites that serve as biomarkers in postmortem serum. Fang et al. [121] selected several metabolites biomarkers related to PMI within 24 h, and proposed that eight metabolites (isoleucine, alanine, proline, valine, etc.) are significant for PMI precision estimation. Chighine et al. [122] utilized the metaboloics method to analyze pericardial fluid to select the PMI biomarkers (hypoxanthine, choline, ethanolamine, and glycine), and found the prediction error in two periods after death. Aqueous humor and vitreous humor are more suitable samples for estimating PMI compared with postmortem serum [123]. Forensic researchers optimized the previous thanatochemistry method, making the metabolomics profile predictive power better than routine potassium ion [124]. Aiello et al. [125] exploited a metabolomics method to analyze the vitreous humor metabolomics profile, which can be used to estimate PMI within the range of 0–96 h. Du T et al. [126] utilizedLC-MS to investigate the postmortem changes in metabolites in femoral muscles in rats. They identified 59 metabolites strongly associated with PMI within 48 h after death. Lu et al. [127] proposed that the multi-organ stacking model also performed well in predicting PMI. Additionally, volatile organic compounds are also considered as potential biomarkers; therefore, GC-MS is also used to separate metabolomics. Sato et al. [128] utilized metabolomics based on GC-MS/MS to analyze 70 metabolites and establish an orthogonal signal correction-partial least squares regression model for predicting PMI within the range of 0–48 h. Richard et al. [129] quantified the content of γ-hydroxybutyrate and its physiological metabolites, γ-aminobutyric acid, 1,4-butanediol, and γ-butyrolactone in various animal tissues using targeted metabolomics based on GC-MS, among which, they disclosed a strong association between the gamma-hydroxybutyrate levels with PMI and the blood and brain of male Sprague–Dawley rats at 15 °C and 20 °C.

### 3.6. Fourier Transform Infrared Spectroscopy (FT-IR)

FT-IR has been a well-used analytical technique for biofluid investigation in recent decades. Due to its non-invasive, observer-independent, and label-free qualities when analyzing biological tissues, samples could undergo morphological detection to verify the changes at the same site of the tissue slides (Table 1). The FT-IR can provide information on various organic molecular substances, including proteins, lipids, nucleic acid, carbohydrates, etc. The characterized peak distribution and relative area response could be confirmed by using curve fitting, which is used to quantify the secondary structure of the target protein. FT-IR is an important adjunct method to the routine histological method in pathology [130]. In forensic science, the FT-IR method has been applied in researching the specific biomarkers of wound age, cause of death in complex cases, and PMI estimation.

FT-IR technology has been used to identify the cause of death in ‘negative autopsies’. Wu et al. [131] used principal component and partial least squares discriminant analysis (PLS-DA) to analyze the FT-IR data, aiding in the identification of deceased who died from DKA. Cheng et al. [132] developed a novel method based on FT-IR, which can ascertain pulmonary fat embolism with a simple approach. Lin et al. [133] used a genetic algorithm-partial least squares regression method to analyze the abnormalities of biofluids (plasma, urine, and saliva) using FT-IR data, and proposed that these biofluid infrared spectro-diagnostic models can help aid in the early identification of diabetic cardiomyopathy. FT-IR was also utilized in cases of heatstroke. Previous studies indicated that the PLS-DA analysis of FT-IR data (plasma, edema fluid in formalin-fixed and paraffin-embedded lung tissues), could help to identify fatal cases of hypothermia/hyperthermia [134,135]. In anaphylactic shock cases, the pulmonary edema fluid is rich in α-helix protein structures and tyrosine-rich proteins compared with other fatal causes of death [136]. FT-IR is a reliable method, which can capture 16 specific absorbance bands in the polarized bronchial epithelia of fatal burn victims [137]. Zhang K et al. [138] utilized FT-IR and PLS-DA analysis to observe the subtle biochemical differences in lung tissue between asphyxia and sudden cardiac death, and proposed that FT-IR technology can help to confirm the cause of death in mildly decomposed corpses.

In postmortem changes, FT-IR shows advantages in estimating PMI, due to its function in obtaining biomedical information from postmortem body fluid. Zhang et al. [139] employed FT-IR to analyze a vitreous humor for estimating PMI. They achieved an impressively low error of 2.018 h within a 48 h postmortem period. And an annular cartilage sample proved to be a good choice for estimating late PMI, which can be detected using FT-IR with a low error of 1.49-day within a 20-day postmortem period [140]. With larger PMI estimation, Baptista et al. [141] utilized FT-IR to analyze the femur and humerus of a human, and proposed that B-type carbonate, A-type carbonates, carbonate/phosphate ratio, crystallinity index, and carbonate ratio related to larger PMI (19–25 year) estimations. Zhang et al. [142] utilized FT-IR technology to identify the muscle hemorrhage in a decomposed rat corpse, which can help to confirm an antemortem wound. In addition, the FT-IR can be used to analyze bone powder from a fracture area, which can help forensic practitioners to judge the antemortem fracture and postmortem fracture [143].

In summary, high-throughput technologies are potential tools for overcoming the challenges of postmortem degradation in forensic practice and studying its patterns. However, postmortem changes are still challenging in forensic science. The single method is still limited in presenting the whole profile in terms of the forensic science question (Table 3). Firstly, high-through technologies still cannot be directly used in forensic practice, other than in the routine approach of protein detection, due to its high cost. Secondly, target biomarkers (accessed from experimental animals) should be validated by further research due to the different proteome expression between human and animals. Thirdly, though proteomics can access massive data about DEPs, the anti-bodies are absent in the market sometimes. However, with the mass spectrometers being purchased in toxicology laboratories, the cost of the proteomics method will cheaper in the future.
ijms-25-01659-t003_Table 3Table 3The main approaches in protein degradation research.MethodAdvantageLimitationMorphological methodSimple, low-consuming, common Not quantitative, false positive in the putrefied tissue ELISAHigh sensitive, high selective, quantitativeSusceptible to interference by putrefaction, poor coverage of target substanceWestern bolt assayHigh sensitive, high selective, quantitativePoor repeatability, complex experimental procedureProteomicsHigh identification accuracy, high throughput, massive dataHigh cost, complex experimental procedure, complex data processingPeptidomics Systematic, high throughput, high sensitive, massive data, less destructive for samplesHigh cost, complex data processingcomplex interpretation [144]Untargeted MetabolomicsSystematic, high throughput, massive metabolites data, massive metabolic pathwayComplex data processing, unquantifiable Targeted metabolomicsAccuracy of quantification and qualificationHigh cost, complex experimental procedure, strict quality control condition, poor coverage of metabolitesFI-TR methodNon-invasive, robustness, low cost, massive data [145]Finickiness in test sample, instrument-dependable, incomplete molecular information in single spectrum [146]


## 4. Discussion

Certain proteins are considered as characteristic biomarkers in the clinical process of some diseases. However, postmortem protein analysis is still limited in forensic pathology, even though the protein changes can help to explain the pathophiosiological mechanism of death in some cases [147]. This review of literature has shown that several novel technologies have been applied in forensic practice, which might be a potent method for handling difficult issues in forensic science (Table 4).

### 4.1. Confirming the Cause of Death

Confirming the cause of death is the main part of forensic pathology, which needs a meticulous process involving systematic autopsy, subsequent morphological examination, toxicological analysis, etc. However, in some cases, defining the cause of death proves challenging due to the absence of morphological changes, such as fatalities related to fatal arrhythmia, hyperkalemia, electrolyte disorder, ketoacidosis, insulin overdose, and heat stroke. In forensic science, a ‘negative autopsy’ is an enigma for forensic pathologists. In clinical medicine, blood biochemical tests can help doctor to assess diseases associated with metabolic disorders. However, blood biochemical tests could not routinely be used to ascertain a cause of death due to the postmortem changes in a corpse, which present a huge difference in biochemical indicators after human death. Researchers are attempting to use protein technologies to solve the difficult cases in forensic practice, such as FT-IR and the metabolomics method (Table 4).

Anaphylactic shock is diagnosed based on classic clinical symptoms and a clear history of exposure to allergens. In forensic practice, although the content of Ig E in postmortem blood can help to diagnose anaphylactic shock, the postmortem hemolysis may affect the Ig E value. Radheshi et al. [148] reported that a woman died from anaphylactic shock due to the unintentional administration of clarithromycin, in which the cause of death was confirmed based on the detection of mast cell tryptase in serum and anti-tryptase in the spleen.

Coronary heart disease is the most common cause of sudden death. Cases of sudden death ignited by acute myocardial ischemia may lack the distinct characteristics in the early stage of acute myocardial infarction. Therefore, confirming death due to acute myocardial infarction relies on several biochemical indicators of myocardial damage in the clinical practice, including troponin, myoglobin, and creatine kinase isoenzymes. However, this method is not infallible due to postmortem changes in the corpse.

With the development of protein techniques, more and more researchers are utilizing these new technologies to help ascertain the cause of death. The researchers have utilized FT-IR in forensic practice to estimate the subtle features of DAI, electric marks, etc. Proteomics and metabolomics can also help to screen more sensitive biomarkers for diseases related to the cause of death. The proteomics method has now been used to detect the toxin composed by peptide and protein, including snake venom and even ricin [149,150]. Graham et al. [151] utilized metabolomics to detect the metabolites of the medulla (brain stem) in sudden infant death syndrome, and confirmed five biomarkers, including ergothioneine, nicotinic acid, succinic acid, adenosine monophosphate, and azelaic acid. Xie et al. [152] proposed a potential metabolic functional network for aconitine-induced malignant arrhythmia by combining proteomics and metabolomics.

### 4.2. Estimation of PMI

The estimation of PMI is a difficult task in forensic science. The relative accuracy of PMI estimation is high in the early stage (<24 h) based on the changes in the corpse, including body temperature, livor mortis, rigor mortis, etc. However, there is still a significant error in estimating late PMI, which has puzzled forensic researchers for a long time.

The chemical changes in cadavers are still considered as one of the most promising directions to study PMI. Proteomics can capture subtle biochemical differences from a large number of proteins, and quantify each protein, making it useful for selecting PMI biomarkers. Hunsucker et al. [153] utilized two-dimensional gel electrophoresis to discover that almost all the proteins in the brain can maintain stable expression for 4 h after the rats were slaughtered. Tavichakorntrakool et al. [154] used the two-dimensional gel electrophoresis to detect the degradation of creatine kinase, myohemoglobin, and heat shock protein 27. In addition, researchers used protein metabolites, such as urea nitrogen and creatinine, as internal standards to correct different causes of death and explore the use of potassium ion in the inference of PMI, and they found that the estimation accuracy of PMI was significantly improved [155].

The development of metabolomics has sparked new interest in inferring PMI by screening the metabolites of cadaver tissue. More and more researchers hope to use novel technologies, particularly in using metabolomics, to evaluate the postmortem chemical changes, for application in PMI estimation. Kaszynski et al. [156] used GC-MS-based metabolomics and identified 17 potential metabolites biomarkers of PMI within 48 h from death. LC-MS-based untargeted metabolomics show an advantage in data acquisition. Du T et al. [126] scanned 16,000 characterized peaks from rat skeletal muscle within 168 of death and selected 59 kinds of biomarkers of PMI using partial least squares. The quantification of putrescine and cadaverine has been used for PMI (within 120 h) estimation, which indicated a significant relationship between these two metabolites and PMI [157].
ijms-25-01659-t004_Table 4Table 4The novel biochemistry technology applied in forensic practice.Forensic IssueForensic ProblemSample Technology Cause of deathInsulin overdoseVitreous humor [81]ProteomicsProtein toxin (snake venom, ricin)Blood plasma [82] Proteomics DAI biomarkers Cerebral tissue [83,84,85]Proteomics DKA biomarkersPulmonary edema fluid [131]FT-IRHeatstrokePlasma, lung tissue, etc. [134,135]FT-IRDiabetic cardiomyopathyBody fluid (plasma, saliva, and urine) [133]FT-IRAsphyxia Lung tissue [138]FT-IRDiabetes and its complicationPostmortem plasma [107]Metabolomics PMI estimationThe attenuated total reflectance in decomposed tissueVitreous humor and annular cartilage [139,140]FT-IRDegradation of proteinSkeletal muscle and skeleton [91,92,93,94,95,96]Proteomics Identify the polypeptideDecomposed body fluid [100,101]Peptidomics Metabolomics profiles of body fluid Plasma, aqueous humor and vitreous humor [124]Metabolomics based on NMR *Violate metabolitesBrain [119,129,157]Metabolomics based on GC-MS * The abbreviation of ‘Nuclear Magnetic Resonance’.


## 5. Conclusions

Proteins play an integral role in the whole life process of organisms. Protein degradation can affect their analysis and utility in postmortem investigations. Complex forensic issues, such as determining the cause of death and estimating late PMI, present great challenges for the job of judicial judgement. The application of novel protein technologies shows promise for addressing complex forensic problems. Although the previous research about postmortem protein degradation is exciting, the mechanism of postmortem protein degradation is difficult to clarify solely through the omics method. And the previous conclusions cannot give a complete explanation for some cases, due to the lack of validations in human corpses. Forensic pathologists are often questioned in court, and further research is needed to clarify the biochemical profiling in postmortem.

Regarding the scientific issue of the forensic scene, it is a difficult task to select characterized biomarkers to clarify the process of postmortem changes. The integration of multi-omics approaches (multidisciplinary methods), especially when combined with the high throughput technologies, provides the potential to overcome the bottlenecks in postmortem studies and solve complex forensic issues. However, considering the difference between experimental animal and human, the potential biomarkers should be verified repeatedly in human cadavers. Forensic pathologists should also notice some biomarkers variation between living human and postmortem corpses and discuss its guiding value in future research.

## Figures and Tables

**Figure 1 ijms-25-01659-f001:**
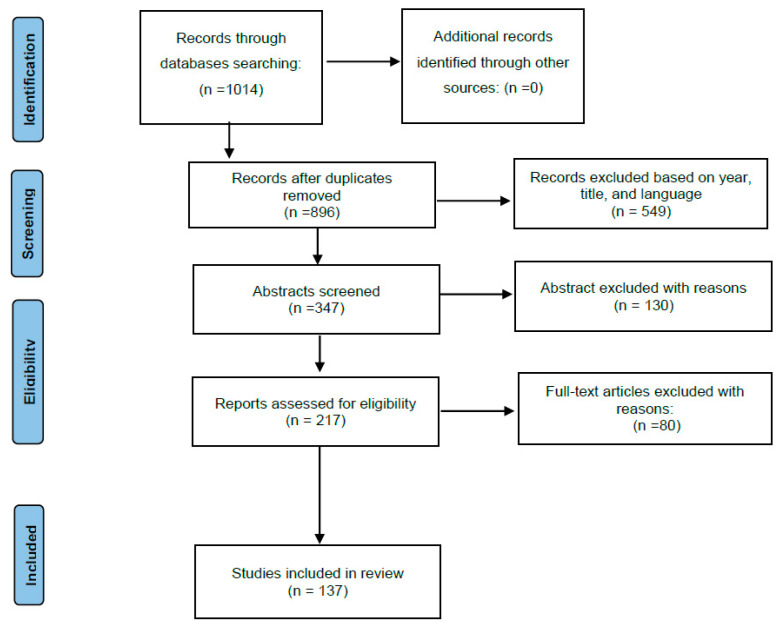
Search method using the PRISMA flowchart.

**Table 1 ijms-25-01659-t001:** The application of IHC assay in forensic science.

Forensic Issue	Forensic Problem	Citation	Target Protein
Cause of death	Snake venom (toxin)	Jacoby-Alner et al. [6]	Notechis venom
	Acute myocardial ischemic	Kondo et al. [7,8,9,10], Mondello et al. [11], Campobasso et al. [12]	Thrombomodulin, von Willebrand factor (VWF), vimentin; CD31; C5b-9, fibronectin; fibronectin, fibrinogen, complement, myoglobi, actin, and desmin;
	Fatal traumatic brain injury	Yang et al. [13], Sieber et al. [14], Maiese et al. [15], Imada et al. [16], Bohnert et al. [17],	Amyloid-β protein, amyloid-β precursor protein, neuron specific enolase (NSE), S100 calcium-binding protein B (S100B), Forkhead Box class O 3a (FOXO3a); CD68, S100; trans-membranous molecule 119 (TMEM119);
	Mechanical asphyxiation	Kim et al. [18], Stemberga et al. [19], Lee et al. [20], Cecchi et al. [21]	Extracellular Signal-Regulated Kinase 2, receptor for advanced glycation end products, aquaporin 5; Surfactant protein-A (SP-A), Hypoxia-induced factor 1-α HIF1α;
	Death due to fire	Doberentz et al. [22]	Heat shock protein 27 (HSP27), HSP70;
	Fatal anaphylaxis and asthma death	Perskvist et al. [23]	Eosinophils (antibodies against human tryptase and chymase)
	Anaphylactic Death:	Esposito et al. [24], Reggiani Bonetti et al. [25]	Tryptase (eosinophilis);
	Heatstroke	Nakagawa et al. [26]	Fibronectin;
	Hypothermia (Wischnewsky spots)	Tsokos et al. [27]; Preuss et al. [28];	Hemoglobin; HSP 70
	Amniotic fluid embolism	Tombolini et al. [29], Wang et al. [30]	CD31, cytokeratins; CK13, CK10/13;
	Drug death	Argo et al. [31], Welte et al. [32], Dettmeyer et al. [33], Cecchi et al. [34]	Caspase 9; myoglobin; leucocytes, T-lymphocytes and macrophages; CD68, Inducible NO synthase (iNOS), CD163, CD15, CD8, CD4, HIF1α;
PMI Estimation	The degeneration of gland	Ortmann et al. [35], De-Giorgio et al. [36]	Pancreatic tissue (insulin and glucagon), thyroid gland (thyreoglobulin and calcitonin); high mobility group box-1 (HMGB1);

**Table 2 ijms-25-01659-t002:** The application of ELISA in forensic science.

Forensic Issue	Forensic Problem	Citation	Target Protein	Samples
Toxicology	Exogenous protein Toxin	Huang et al. [51]	Snake venom	Multi-organ
	Cardiotoxicity	Sharif et al. [52]	S100β	Serum
	Neurotoxicity	Xiong et al. [53]; Wei et al. [54]	Tumor necrosis facorα (TNFα); interleukin-1β (IL-1β), IL-6,	Serum
Postmortem detection of the cause of death	Traumatic brain injury	Li et al. [55], Olczak et al. [56,57,58]	S100β; microtubule associated protein tau (MAPT), tau, glial fibrillary acidic protein (GAFP); GAFP, MAPT, S100β, Spectrin Alpha, Non-Erythrocytic 1 (SPTAN1), pro-Brain-Derived Neurotrophic Factor (pro-BDNF);	Serum and urine
	Ischemic sudden cardiac death	Meng et al. [59], Sapouna et al. [60] Yu et al. [61]	heart fatty acid binding protein (H-FABP); cardiac troponin I c(TnI); vesicular integral membrane protein 36 (LMAN2), Calpain 1 (CAPN-1), valosin-containing protein;	Serum
	Acute kidney injury	Keltanen et al. [62]	Cystatin C, neutrophil gelatinase-associated lipocalin;	Urine
	Sudden infant death syndrome(SIDS)	Haynes et al. [63]	5-hydroxytryptamine	Serum
PMI estimation	The degeneration of protein in body fluid	Kasuda et al. [64], Zhang et al. [65]	VWF; Vascular endothelial growth factor (VEGF)/HIF-1α	Serum

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
