# Peer review of "The Role of Protein Degradation in Estimation Postmortem Interval and Confirmation of Cause of Death in Forensic Pathology: A Literature Review"

_ijms, 2024, doi:10.3390/ijms25031659_

Round 1

Reviewer 1 Report

Comments and Suggestions for Authors

 The authors review the postmortem application of proteins in forensic pathology. The aim of the study is interesting given the large increase in research and practical application in this field. To address this objective the authors conduct a narrative review of publications in this field. The main problem with this work is the absence of methodology. No information is provided on the criteria with which the literature review was carried out.

Narrative reviews, according to the according to the hierarchy of evidence, are found at the bottom of the pyramid (exposed are at the bottom of the pyramid (exposed to a high risk of bias to the possibility of presenting a high risk of bias, mainly because of their subjectivity and lack of methodology).

As an example, section “2.1. Immunohistochemistry (IHC) assay” includes only 4 references on the subject. Immunohistochemistry (IHC) is an important diagnostic tool in anatomic and surgical pathology but is used less frequently in forensic pathology. Degradation of tissue because of postmortem decomposition is believed to be a major limiting factor, although it is unclear what impact such degradation actually has on IHC staining validity. However, there are numerous publications on the subject in specialized journals and by authors recognized in the scientific field. As an example, I include some references, in addition to which systematic reviews have also been published, which I add.

Maeda, H., Fujita, M. Q., Zhu, B. L., Ishida, K., Quan, L., Oritani, S., & Taniguchi, M. (2003). Pulmonary surfactant-associated protein A as a marker of respiratory distress in forensic pathology: assessment of the immunohistochemical and biochemical findings. Legal Medicine5, S318-S321.

Li, D. R., Zhu, B. L., Ishikawa, T., Zhao, D., Michiue, T., & Maeda, H. (2006). mmunohistochemical distribution of S-100 protein in the cerebral cortex with regard to the cause of death in forensic autopsy. Legal medicine, 8(2), 78-85.

Li, D. R., Zhu, B. L., Ishikawa, T., Zhao, D., Michiue, T., & Maeda, H. (2006). Immunohistochemical distribution of S-100 protein in the cerebral cortex with regard to the cause of death in forensic autopsy. Legal medicine, 8(2), 78-85.

Barranco, R., Bonsignore, A., & Ventura, F. (2021). Immunohistochemistry in postmortem diagnosis of acute cerebral hypoxia and ischemia: A systematic review. Medicine, 100(25).

Nishi, K., Tanegashima, A., Yamamoto, Y., Ushiyama, I., Ikemoto, K., Yamasaki, S., ... & Brinkmann, B. (2003). Utilization of lectin-histochemistry in forensic neuropathology: lectin staining provides useful information for postmortem diagnosis in forensic neuropathology. Legal medicine, 5(3), 117-131.

De-Giorgio, F., Bergamin, E., Baldi, A., Gatta, R., & Pascali, V. L. (2023). Immunohistochemical expression of HMGB1 and related proteins in the skin as a possible tool for determining post-mortem interval: a preclinical study. Forensic Science, Medicine and Pathology, 1-17.

Mondello, C., Cardia, L., & Ventura-Spagnolo, E. (2017). Immunohistochemical detection of early myocardial infarction: a systematic review. International journal of legal medicine, 131, 411-421.

Barranco, R., Bonsignore, A., & Ventura, F. (2021). Immunohistochemistry in postmortem diagnosis of acute cerebral hypoxia and ischemia: A systematic review. Medicine, 100(25).

Zissler, A., Stoiber, W., Steinbacher, P., Geissenberger, J., Monticelli, F. C., & Pittner, S. (2020). Postmortem protein degradation as a tool to estimate the PMI: A systematic review. Diagnostics, 10(12), 1014.

Wang, Q., Michiue, T., & Maeda, H. (2012). Immunohistochemistry of neuronal apoptosis in fatal traumas: the contribution of forensic molecular pathology in medical science. IntechOpen.

Similar situations for the rest of the sections.

Therefore, in my opinion, believe that the paper under these conditions should be rejected. I encourage the authors to rethink this objective by applying a methodology that allows obtaining sufficient scientific evidence to be a reference in this area and that can be reproducible.

Comments on the Quality of English Language

I believe that a minor revision of the English language is necessary as well as a proofreading of typos. 

Author Response

Dear reviewers:

Thanks for your comments concerning our review entitled " The role of protein degradation in estimation Postmortem interval and confirmation of cause of death in forensic pathology: A Literature Review " (ijms-2791991).

The comment is very helpful for revising and improving our paper. We have considered comment carefully and have made correction. The main corrections in the paper and the responds to the reviewer’s comments are as follows:

Responds to the reviewer’s comments:

Reviewer #1

  1. The main problem with this work is the absence of methodology. No information is provided on the criteria with which the literature review was carried out.

Responds: Thanks for your reminding. We have added methodology in line 77-86 on page 2 and 3. And we drew a flowchart about our searching method by PRISMA.

  1. 2. As an example, section “2.1. Immunohistochemistry (IHC) assay” includes only 4 references on the subject. Immunohistochemistry (IHC) is an important diagnostic tool in anatomic and surgical pathology but is used less frequently in forensic pathology. Degradation of tissue because of postmortem decomposition is believed to be a major limiting factor, although it is unclear what impact such degradation actually has on IHC staining validity. However, there are numerous publications on the subject in specialized journals and by authors recognized in the scientific field. As an example, I include some references, in addition to which systematic reviews have also been published, which I add. Similar situations for the rest of the sections.

Responds: Thanks for your comments. In the following, we address these concerns one by one.

(1) We reviewed the all of the papers searched by the new methodology (including your added papers), and revised the part of Immunohistochemistry (IHC) in the latest revised manuscript.

(2) Each part contents are revised after we reviewed the relevant papers searched with the new methodology.

  1. I encourage the authors to rethink this objective by applying a methodology that allows obtaining sufficient scientific evidence to be a reference in this area and that can be reproducible

Responds: Thanks very much for your reminding. We established a methodology to avoid the bias risks. However, the searching methodology contains several terms, due to multiple technologies involved.

  1. I believe that a minor revision of the English language is necessary as well as a proofreading of typos

Responds: Thanks very much for your reminding. We revised the manuscript sentence by sentence carefully. However, the manuscript wasn’t proofread by professional editing service due to the tight deadline. Please inform me if the language of manuscript need another proof. And I’d be grateful if you can recommend several professional editing services.  

Reviewer 2 Report

Comments and Suggestions for Authors

This is a useful review in the context of the application of proteins in the postmortem. Personally I think the text is a bit chaotic in some parts. In particular, I suggest removing introductory parts in any paragraph where obvious information is mentioned (we all know what a protein is and what mass spectrometry is!). Therefore, I would suggest adding more technical information on the application of post-mortem protein studies with a greater number of studies extrapolated from the literature. Furthermore, the authors should also evidence the materials and methods (which search engine has been used for the review) and should suggest what future applications of this discipline could be in practice. To this end I suggest inserting an image. Finally, I recommend considering the following manuscripts:

-Exploring the Potential of Proteome Analysis as a Promising Tool for Evaluation of Sudden Cardiac Death (SCD) in Forensic Settings: A Literature Review. Int J Mol Sci. 2023 Sep 20;24(18):14351.

-Systematic Review on Post-Mortem Protein Alterations: Analysis of Experimental Models and Evaluation of Potential Biomarkers of Time of Death. Diagnostics (Basel). 2022 Jun 17;12(6):1490. doi: 10.3390/diagnostics12061490. PMID: 35741301; PMCID: PMC9222196.

Comments on the Quality of English Language

Good

Author Response

Dear reviewers:

Thanks for your comments concerning our review entitled " The role of protein degradation in estimation Post-mortem interval and confirmation of cause of death in forensic pathology: A Literature Review " (ijms-2791991).

The comment is very helpful for revising and improving our paper. We have considered comment carefully and have made correction. The main corrections in the paper and the responds to the reviewer’s comments are as follows:

Responds to the reviewer’s comments:

  1. This is a useful review in the context of the application of proteins in the postmortem. Personally I think the text is a bit chaotic in some parts. In particular, I suggest removing introductory parts in any paragraph where obvious information is mentioned (we all know what a protein is and what mass spectrometry is!). Therefore, I would suggest adding more technical information on the application of post-mortem protein studies with a greater number of studies extrapolated from the literature.

Respond: Thanks very much for your comments. In the following, we address these concerns one by one.

(1) We have deleted introductory parts of techniques, which seems like a bit chaotic in manuscript. And we revised the manuscript sentence by sentence carefully.

(2) We have added more technical information after review a greater number of studies, which including a total of 137 papers about postmortem protein and its metabolits detection. 

  1. Furthermore, the authors should also evidence the materials and methods (which search engine has been used for the review) and should suggest what future applications of this discipline could be in practice.

Respond: Thanks for your reminding. We have added methodology in line 77-86 on page 2 and 3. And we drew a flowchart about our searching method by PRISMA.

  1. Thanks for your reminding. We have added methodology in line 77-86on page 2 and 3.And we drew a flowchart about our searching method by PRISMA.

Respond: Thanks for your reminding. We have inserted the flowchart as Figure 1 on page 3.

Reviewer 3 Report

Comments and Suggestions for Authors

The article is of great interest to forensic pathologists, and it's a lovely narrative review.

It would be nice if the authors had tried to make a systematic review, but I understand that for forensic science, especially forensic pathology, is uneasy.

I think that that article could be enriched with an explanation of the problems related to this kind of research and maybe with some more details about using IHC in postmortem. 

Also, the unsolvable problem of validation of the methods needs to be mentioned. 

Author Response

Dear reviewers:

Thanks for your comments concerning our review entitled " The role of protein degradation in estimation Postmortem interval and confirmation of cause of death in forensic pathology: A Literature Review " (ijms-2791991).

The comment is very helpful for revising and improving our paper. We have considered comment carefully and have made correction. The main corrections in the paper and the responds to the reviewer’s comments are as follows:

Responds to the reviewer’s comments:

  1. The article is of great interest to forensic pathologists, and it's a lovely narrative review. It would be nice if the authors had tried to make a systematic review, but I understand that for forensic science, especially forensic pathology, is uneasy.

Respond: Thanks for your reminding. We have added methodology in line 77-86on page 2 and 3. And we drew a flowchart about our searching method by PRISMA.

  1. I think that that article could be enriched with an explanation of the problems related to this kind of research and maybe with some more details about using IHC in postmortem.

Respond: Thanks for your comments. We reviewed the all of the papers searched by the new methodology (including your added papers), and added more details about IHC in postmortem.

  1. Also, the unsolvable problem of validation of the methods needs to be mentioned.

Respond: Thanks for your comments. We have added the unsolvable problem of validation of the methods in line 460 - 470 on page 12.
